# HDAC Inhibitors Exert Anti-Myeloma Effects through Multiple Modes of Action

**DOI:** 10.3390/cancers11040475

**Published:** 2019-04-04

**Authors:** Yoichi Imai, Mitsuhito Hirano, Masayuki Kobayashi, Muneyoshi Futami, Arinobu Tojo

**Affiliations:** 1Department of Hematology/Oncology, Research Hospital, Institute of Medical Science, University of Tokyo, Tokyo 108-8639, Japan; a-tojo@ims.u-tokyo.ac.jp; 2Division of Molecular Therapy, Advanced Clinical Research Center, The Institute of Medical Science, University of Tokyo, Tokyo 108-8639, Japan; m-hirano@ims.u-tokyo.ac.jp (M.H.); masa-k@ims.u-tokyo.ac.jp (M.K.); m-futami@umin.ac.jp (M.F.)

**Keywords:** HDAC inhibitor, histone modification, multiple myeloma, panobinostat, HDAC6, HSP90

## Abstract

HDACs are critical regulators of gene expression that function through histone modification. Non-histone proteins and histones are targeted by these proteins and the inhibition of HDACs results in various biological effects. Moreover, the aberrant expression and function of these proteins is thought to be related to the pathogenesis of multiple myeloma (MM) and several inhibitors have been introduced or clinically tested. Panobinostat, a pan-HDAC inhibitor, in combination with a proteasome inhibitor and dexamethasone has improved survival in relapsing/refractory MM patients. We revealed that panobinostat inhibits MM cell growth by degrading the protein PPP3CA, a catalytic subunit of calcineurin. This degradation was suggested to be mediated by suppression of the chaperone function of HSP90 due to HDAC6 inhibition. Cytotoxicity due to the epigenetic regulation of tumor-associated genes by HDAC inhibitors has also been reported. In addition, HDAC6 inhibition enhances tumor immunity and has been suggested to strengthen the cytotoxic effects of therapeutic antibodies against myeloma. Furthermore, therapeutic strategies to enhance the anti-myeloma effects of HDAC inhibitors through the addition of other agents has been intensely evaluated. Thus, the treatment of patients with MM using HDAC inhibitors is promising as these drugs exert their effects through multiple modes of action.

## 1. Introduction

Multiple myeloma (MM) is a B-cell malignancy that can be difficult to treat. The introduction of autologous stem cell transplantation and other novel drugs including a proteasome inhibitor (bortezomib) and IMiDs (thalidomide, lenalidomide, and pomalidomide), has improved the survival rates of patients with MM, however, many patients with relapsing/refractory MM remain and new therapies to treat such individuals are needed.

HDACs are known to be involved in many biological processes related to malignancy such as apoptosis, senescence, differentiation, and angiogenesis. Furthermore, aberrant expression of these proteins in several types of human tumors including gastric, colorectal, liver, breast, and lung cancer has been reported. In MM, progression-free survival was significantly shorter in myeloma patients with higher levels of class I HDAC expression [1]. For this reason, HDACs have been proposed to be good candidates for the targeted treatment of MM.

Human HDACs can be divided into 11 subtypes based on four classes (I, IIa, IIb, and IV) [2]. Differences in the subcellular locations, substrate specificities, tissue expression patterns, and enzymatic activities of each class of HDACs confer specific biological actions to each one. Classes, substrates, biological functions, and IC_50_ values for panobinostat, ricolinostat (an HDAC6-selective inhibitor), and vorinostat (a pan-HDAC inhibitor) toward the main HDACs are displayed in Table 1.

Class I HDACs (1, 2, 3, and 8) are generally located in the nucleus where they regulate gene expression through the deacetylation of histones. Thus, these have been suggested to function as epigenetic regulators through histone modifications. Epigenetic mechanisms are essential for the homeostatic expression of tissue-specific genes and disrupting this balance can result in malignant cellular transformation due to altered gene functions. However, the fact that aberrant epigenetic mechanisms are reversible suggests the possibility of epigenetic therapy [5]. Based on this, epigenetic regulation seems to be a good target of HDAC inhibitors for the treatment of patients with MM.

Although class II HDACs (4, 5, 6, 7, 9, and 10) can shuttle between the nucleus and the cytoplasm, substrates of HDAC6 are unique among class II HDACs. The main substrates of this enzyme are tubulin and HSP90, and HDAC6 regulates protein stability and cell signaling pathways via interactions with both proteins. Targeting HDAC6 can result in anti-tumor effects that are induced by the degradation of proteins and the inhibition of signaling pathways related to oncogenesis.

The immunogenicity-like ligand expression of immunoreceptors on natural killer cells is also regulated by HDACs [6]. Here, we describe the clinical use and mechanisms of action, including epigenetic, protein stabilizing, and immunogenic, of HDAC inhibitors with respect to the treatment of MM.

## 2. HDAC Inhibitors in the Clinic

Several HDAC inhibitors are used in the clinic. Vorinostat and romidepsin (class I HDAC inhibitor) are approved for the treatment of refractory cutaneous and peripheral T cell lymphoma [2]. Panobinostat (a pan-HDAC inhibitor) in combination with bortezomib has been approved for relapse/refractory MM patients. PANORAMA 1, a phase III trial involving panobinostat plus bortezomib and dexamethasone versus placebo plus bortezomib and dexamethasone, was carried out in patients with relapse/refractory MM [7]. The rate of significant response, including complete response and near complete response, was markedly higher in the panobinostat group than in the placebo group. Moreover, the panobinostat group experienced improved progression-free survival. Thrombocytopenia, diarrhea, asthenia, and fatigue were the main grade 3 and 4 drug-related adverse events. Combined panobinostat and bortezomib and dexamethasone therapy is thus expected to be effective even in patients who are resistant to bortezomib [8].

## 3. Roles of Each HDAC in the Treatment of MM (Expression, Prognostic Significance, Molecular Mechanisms and Therapeutic Targets Revealed by Preclinical Studies)

### 3.1. HDAC1 and HDAC3 (Class I)

As described previously, high expression of several HDACs is related to poor prognosis for MM patients [1]. An analysis of HDAC1 expression based on the immunohistochemical staining of bone marrow samples from transplant-eligible MM patients revealed a significant correlation between high levels of myeloma cell HDAC1 expression and poor prognosis. There was also a significant increase in the expression of HDAC3 in myeloma cells when compared to that in normal plasma cells. Together, these results indicate that the overexpression of class I HDACs, and particularly HDAC1, is associated with poor prognosis in MM.

Moreover, in this disease, many patients suffer from osteolytic lesions, partly due to the suppression of osteoblast differentiation. Activation of the RUNX2/CBFA1/AML3 transcription factor axis is essential for osteoblast differentiation and RUNX2 activity in osteoblast precursors is transcriptionally inhibited in MM [9]. In myeloma-exposed pre-osteoblasts, GFI1 binds the *RUNX2* promoter and gene suppression is maintained through the recruitment of histone modifiers including HDAC1 and EZH2 [10]. Accordingly treating pre-osteoblasts from MM patients with EZH2 or HDAC1 inhibitors was found to reverse the repressive chromatin architecture at *RUNX2* and induce differentiation to osteoblasts [11].

Epigenetic regulation is one of the main biological functions of HDACs including class I HDACs (Table 1). The patterns of DNA methylation and posttranslational modifications of histone regulate the epigenome [12]. Acetylation of histones and methylation at the lysine 9 residue of histone 3 lead to activated gene expression. In contrast, the suppression of gene expression is associated with histone deacetylation by HDACs, which is frequently associated with regions of DNA methylation. Inappropriate silencing of tumor suppressor genes might be related to the occurrence of various type of cancers. HDAC inhibitors cause the accumulation of acetylated histones in nucleosomes. The hydroxamic acid parts of HDAC inhibitors bind to the zinc in the tubular pocket of HDACs and this interaction was suggested to inhibit the catalytic activity of HDACs. The efficacy of HDAC inhibitors as epigenetic modulators of histone modification for the treatment of MM has been suggested by several preclinical studies [13,14]. Vorinostat, a class I/II HDAC inhibitor, was shown to modify the acetylation and methylation of core histones and tightly restrict enzyme accessibility at the *p21^WAF1^* promoter region of myeloma cells [15]. This epigenetic modulation was suggested to induce the expression of p21^WAF1^, a target of proteasome inhibition that is stabilized by bortezomib, and its induction is related to apoptosis in myeloma cells [16]. Furthermore, pre-treatment with bortezomib enhances oxidative injury and apoptosis induced by vorinostat in MM cells. Indeed, this combination was found to be effective for chemotherapy-resistant MM cells [14].

Although *RAS* is frequently mutated in MM patients and known to drive disease progression, it also mediates growth inhibitory effects and apoptosis through activation of the tumor-suppressive RASSF. Expression of RASSF4 is downregulated during MM disease progression and its low expression is related to poor prognosis [17]. Further, the overexpression of RASSF4 was found to reduce primary myeloma cell viability and block tumor growth in the murine 5T3MM model. Linking RAS to proapoptotic pathways was suggested to be one molecular mechanism associated with the role of RASFF4 as a tumor suppressor. This protein interacts with the mammalian sterile 20-like kinases MST1 and MST2 and these interactions lead to a clear increase in the phosphorylation of SAPK/JNK, c-Jun, p38, and p53, which are related to apoptosis. Moreover, treating myeloma cells with quisinostat, an HDAC inhibitor, was found to increase *RASSF4* mRNA expression and this overexpression significantly increased the sensitivity of myeloma cells to bortezomib. These results provide rationale for exploiting the epigenetic upregulation of RASSF4 using HDAC inhibitors for the treatment of patients with tumors displaying low *RASSF4* expression.

In MM, the histone methyltransferase EZH2 is aberrantly activated. EZH2 regulates cell proliferation in hematopoietic cells; as such, treating a large panel of myeloma cell lines with specific inhibitors of this protein leads to ubiquitous global H3K27 demethylation [18]. However, sensitivity to a single agent was observed in only a subset of cell lines. In contrast, combining EZH2 and HDAC inhibitors epigenetically perturbed oncogenic pathways and signaling and resulted in enhanced anti-myeloma effects.

### 3.2. HDAC4 (Class IIa)

There is no significant difference in HDAC4 expression between normal plasma cells and myeloma cell lines [1]. Epi-miRNAs comprise a subclass of tumor suppressor miRNAs that facilitate the reversion of epigenetic aberrations through the downregulation of HDACs. miR-29b is a well-established epi-miRNA that is related to the reactivation of promoter-hypermethylated tumor suppressor genes like p15 (INK4b) and ESR1through the downregulation of DNMT expression [19]. It was shown that miR-29b specifically targets HDAC4 and that this protein is involved in the pathogenesis of myeloma. In fact, silencing HDAC4 with shRNAs was found to inhibit MM cell survival and migration and trigger apoptosis and autophagy, along with the induction of miR-29b expression via promoter hyperacetylation, leading to the downregulation of pro-survival miR-29b targets (SP1 and MCL-1). Moreover, treatment with vorinostat was found to upregulate miR-29b, overcoming the negative control exerted by HDAC4. Importantly, overexpression or inhibition of miR-29b, respectively, potentiated or antagonized vorinostat activity in MM cells, as shown in vivo by the observed strong synergism between miR-29b synthetic mimics and vorinosta in a murine xenograft model of human MM. Stamato et al. provided novel insights into the epigenetic regulation of miR-29b suppression in MM [20]. In plasma cells derived from patients with MM, the correlation between miR-29b and EZH2 mRNA expression was inverse. Further, inhibiting EZH2 leads to miR-29b upregulation through reduced H3K27-trimethylation (H3K27me3) of its promoter regions. The induction of miR-29b through EZH2 inhibition was also found to downregulate major miR-29b pro-survival targets, such as SP1, MCL-1 and CDK6.

### 3.3. HDAC6 and HDAC10 (Class IIb)

Class IIb HDACs are composed of HDAC6 and HDAC10. Although there is no significant difference in HDAC6 expression between normal plasma cells and myeloma cell lines, the expression of HDAC10 was found to increase in myeloma cells when compared to that in normal plasma cells [1]. Multiple and abundant proteins including immunoglobulins are produced by MM cells and some of them are misfolded or unfolded [21]. These proteins are sometimes cytotoxic; thus, proteasomes contribute to the maintenance of protein homeostasis by degrading those ubiquitinated misfolded and unfolded proteins [22]. Accordingly, myeloma cells are more sensitive to proteasome inhibition than normal cells due to their dependence on proteasomes for the clearance of these cytotoxic proteins. Proteasome inhibitors can exert cytotoxicity against myeloma cells partly through the deposition of misfolded/unfolded proteins. An aggresome is a pericentriolar microtubule-based structure and aggresome formation generally occurs as a cellular response to the excessive accumulation of misfolded/unfolded proteins [23,24]. Bortezomib is a reversible inhibitor of chymotryptic activity that functions by targeting the 20S subunit of proteasomes and is remarkably effective for treating MM [25,26]. The treatment of MM cells with bortezomib also induces aggresome formation [27]. Aberrant degradation of accumulated misfolded/unfolded proteins has been suggested to lead to the resistance of myeloma cells to proteasome inhibitors. HDAC6 is essential for aggresome formation, which occurs through the recruitment of misfolded/unfolded proteins to dynein motors for transport to aggresomes; accordingly, HDAC6 inhibition can result in failed protein aggregate clearance. Panobinostat and its combination with bortezomib can synergistically induce apoptosis by inhibiting protein degradation and the massive accumulation of polyubiquitinated proteins by targeting both proteasomal and aggresomal protein degradation systems (Figure 1) [28]. This synergistic activity of HDAC inhibitors with bortezomib is mainly due to HDAC6 and selective HDAC6 inhibitors like ricolinostat and WT161 might reduce toxicity related to the off-target effects of pan-HDAC inhibitors [3,29].

The transcription factor Sp1 is overexpressed in myeloma cells and targets the pro-survival factors IRF4 and c-Myc [30]. The treatment of myeloma cells with panobinostat was found to activate caspase-8 via Sp1 reduction. However, the combination therapy of panobinostat with bortezomib or carfilzomib showed synergistic anti-myeloma effects, whereas addition of the caspase-8 inhibitor z-IETD-FMK abolished the Sp1 downregulation induced by this combination. It was thus suggested that caspase-8-mediated post-translational Sp1 degradation plays an important role in the synergistic anti-myeloma effects of panobinostat and proteasome inhibitors in combination.

We previously attempted to find a novel HDAC target for the treatment of MM and PPP3CA, a catalytic subunit of calcineurin, was revealed to be a candidate [31]. PPP3CA dephosphorylates NFATc1 as a serine/threonine protein phosphatase and regulates its translocation from the cytoplasm to the nucleus. This translocation is indispensable for T-cell activation, and immunosuppressive agents including FK506 and cyclosporine A block this process by inhibiting the interaction between PPP3CA and its heterodimeric partner calcineurin B. Defective B-cell activation induced by calcineurin inactivation suggests the important role of calcineurin in lymphocyte-function [32]. Regarding the relevance of its aberrant activation to carcinogenesis, mouse models of T-cell acute lymphoblastic leukemia due to the aberrant activation of calcineurin have been reported [33].

In MM, we found high *PPP3CA* expression in myeloma cells from patients with advanced disease and a possible correlation between this event and disease pathogenesis was suggested. Resistance to bortezomib-containing chemotherapies was also demonstrated in patients with high *PPP3CA* expression. Further, the knock down of PPP3CA in myeloma cells was shown to reduce MM cell growth maintained via NF-κB signaling. Moreover, either panobinostat or ricolinostat was found to reduce PPP3CA expression through protein degradation in myeloma cells. Results indicated that PPP3CA functions as a target protein of HSP90 in MM cells and that treatment with HDAC inhibitors can result in the release of PPP3CA from over-acetylated HSP90 via HDAC6 inhibition (Figure 2) [3]. Thus, HDAC inhibitors might block the chaperone function of HSP90 and induce the degradation of PPP3CA [31]. Panobinostat treatment reduces MM cell growth and its combined use with FK506 was found to enhance the downregulation of PPP3CA and cell growth induced by panobinostat. This combination was suggested to lead to the diminished protection of PPP3CA from protein degradation [34].

Combination therapy comprising bortezomib and an HSP90 inhibitor has shown promising anti-myeloma effects [35]. However, 17-AAG, an HSP90 inhibitor, increases the expression of HSP70, which protects cancer cells from apoptosis; this was thought to reduce its anti-myeloma effects [36]. Increases in HSP70 expression by panobinostat were found to be slight and the loss of anti-myeloma effects due to HSP70 expression were less pronounced with bortezomib and panobinostat than with bortezomib and 17-AAG [31].

Although HDAC10 is the second member of class IIb, less is known about its relationship with carcinogenesis as compared to that with HDAC6. No significant difference was found in HDAC10 expression between normal plasma cells and myeloma cell lines [1]. HDAC10 promotes autophagy-mediated survival and its inhibition was suggested to lead to the sensitization of tumor cells to cytotoxic drug [37]. This protective effect was found to be mediated by autophagy via interactions with HSP70 family proteins.

### 3.4. HDAC11 (Class IV)

HDAC11 is the smallest HDAC as well as only one member of class IV HDAC [38]. Previously, no significant difference was found for HDAC11 expression between normal plasma cells and myeloma cell lines [1]. As one of its function, much attention has been paid to immune-state modification [39]. APCs (antigen-presenting cells) induce both the activation and tolerance of T-cells and HDAC11 was suggested to regulate this switch from immune activation to tolerance. HDAC11 is an epigenetic negative regulator of IL-10, which keeps immune responses in check, preventing self-tissue damage [40]. Thus, HDAC11 inhibition impairs antigen-specific T-cell responses through the upregulation of IL-10. HDAC11 is also a key epigenetic regulator of IL-10 in myeloid cells and inhibits the expansion and function of MDSCs (myeloid-derived suppressor cells) [41]. MDSCs are derived from immature myeloid cells and function as suppressors of anti-tumor T-cells in the tumor microenvironment [41]. In a xenograft model of tumor-bearing mice, the growth of tumors was enhanced in HDAC11-KO mice compared to that in WT (wild-type) mice. Accordingly, the inhibition of HDAC11 might impair efficacious immunotherapies.

## 4. Strategy for Enhancing the Anti-Myeloma Effect of HDAC Inhibitors by Adding Other Agents

### 4.1. IMiDs

The combination treatment of lenalidomide and the class-I HDAC-selective inhibitor entinostat induces synergic cytotoxicity, partly through the downregulation of c-Myc with decreased levels of CRBN, a primary target protein of IMiDs [42]. CRBN is an IMiDs-binding protein and the substrate adaptor of the CRL4^CRBN^ E3 ubiquitin ligase [43,44]. IMiDs induce the recruitment of specific substrates including IKZF1/3 to E3 ubiquitin ligase and display anti-myeloma effects through the ubiquitination and subsequent proteasomal degradation of IKZF1/3. This decrease in CRBN is supported by results indicating that the sequential treatment of MM cells with entinostat followed by lenalidomide is less effective than simultaneous treatment with these agents. Combination therapy including ricolinostat and lenalidomide was also found to exert synergistic cytotoxicity against myeloma cells in the absence of altered CRBN expression. Further, the acetylation of histone H3K9 and activation of caspase-3, induced by panobinostat, were shown to be inversely correlated with the reduction in HO-1/IRF4/MYC protein levels [45]. Lenalidomide stabilizes CRBN and facilitates IRF4 degradation in MM cells and the combination of panobinostat and lenalidomide was found to exert synergistic effects partly due to the simultaneous suppression of HO-1, IRF4, and c-Myc.

### 4.2. Cladribine (2CdA)

The combination of entinostat, a selective class I HDAC inhibitor, and 2CdA synergistically inhibits the proliferation of several myeloma cell lines [46]. This combination therapy leads to cell cycle G1 arrest accompanied by decreases in Cyclin D1 and E2F-1 expression and increases in p21^waf−1^. Apoptosis and the DNA damage response, as evidenced by the enhanced phosphorylation of H2A.X and Chk2, was also observed.

### 4.3. PI3K Inhibitor

In MM, the interaction between myeloma cells and the bone marrow environment sustains proliferation of the former cell type, which is driven by genetic alterations. The PI3K/mTOR/AKT pathway is an integral regulator of this signaling network and its activation is related to survival and drug resistance in MM. VS-5584 is a synthesized purine analog and an equally potent inhibitor of both PI3K (all four isoforms of catalytic subunit p110 (α, β, γ, δ)) and mTORC1 and 2 [47]. VS-5584 effectively inhibits the growth of MM cell lines even in the presence of IL-6 and IGF-1 and this inhibition is partially dependent on Bim. The tumor suppressor RARRES3 is downregulated with the progression of B-CLL, a mature B-cell malignancy [48]. Treating myeloma cells with VS-5584 was found to upregulate RARRES3 expression [47]. Furthermore, VS-5584 exerts synergistic anti-myeloma effects with the HDAC inhibitor panobinostat. Thus, combination therapy with VS-5584 and an HDAC inhibitor was suggested to be a promising novel clinical strategy for relapsing/refractory myeloma.

### 4.4. Dual Inhibition of HDAC and BCL-XL

HDAC inhibitors have been clinically approved for the treatment of cutaneous T-cell lymphoma and multiple myeloma. HDAC inhibitors also display clinical activity against acute myelogenous leukemia, non–small cell lung cancer, and estrogen receptor-positive breast cancer. Several clinical trials using HDAC inhibitors, including combination therapies with immunomodulatory agents, are currently underway. However, predicting the sensitivity of each cancer is difficult and reliable strategies to facilitate this are thus needed. Using cell lines originating from diverse tumor types, the immediate HDAC inhibitor-mediated induction of genes related to the apoptotic pathway was examined. As a result, it was revealed that ATF3 is immediately induced by HDAC inhibition and that it is required for apoptosis [49]. This protein functions as a proapoptotic factor through direct transcriptional repression of the pro-survival factor BCL-XL. These findings provide the rationale for the dual inhibition of HDAC and BCL-XL to cooperatively overcome inherent resistance to HDAC inhibitors across diverse tumor cell types.

### 4.5. mTOR Inhibitors

The efficacy of suppressing tumor growth with combination therapy comprising an mTOR inhibitor (rapamycin) and HDAC inhibitor (entinostat) has been reported using more than 60 cell lines derived from human cancer [50]. It was revealed that the Myc/E2F axis is significantly inhibited by this drug combination in several myeloma cell lines. Although the combination inhibited both *E2F1* mRNA and protein expression, it decreased Myc protein levels and slightly increased *Myc* mRNA expression. These results were explained by the increased degradation of Myc, which occurred with this drug combination, and the fact that cells with *Myc* mutations were refractory to this combination.

### 4.6. 5-Azacytidine (AZA)

The genetic silencing or pharmacological inhibition of HDAC3 leads to the suppression of myeloma cell proliferation. This was found to be mediated by the downregulation of DNMT1 by HDAC3, but not HDAC1 or HDAC2 [51]. DNMTs methylates DNA at the carbon-5 position of cytosines mainly within CpG sites as an epigenetic process involved in the regulation of gene expression [52]. Preferential methylation by DNMT1 is performed at hemi-methylated CpG sites to maintain DNA methylation during DNA replication and cell division. C-Myc is a transcriptional regulator of DNMT1 and is degraded after the hyperacetylation induced by HDAC3 inhibition. Thus, DNMT1 is epigenetically downregulated through HDAC3 inhibition. Moreover, HDAC3 inhibition results in the hyperacetylation of DNMT1 and a reduction in the stability of the DNMT1 protein. Combined HDAC3 inhibitor and AZA, a DNMT1 inhibitor, was found to synergistically downregulate DNMT1, resulting in the growth inhibition and apoptosis of MM cell lines and myeloma patient-derived cells [51]. The efficacy of this combination treatment was confirmed in vivo using a murine xenograft MM model and the rationale for combined treatment using HDAC3 and DNMT1 inhibitors, as a novel strategy for myeloma therapy, was suggested. The described strategy for enhancing the anti-myeloma effect of HDAC inhibitors by adding other agents is summarized in Table 2.

## 5. Modification of Immune Functions by HDAC Inhibitors for the Treatment of Myeloma

The proliferation of myeloma cells in vivo has been suggested to be regulated by the state of immunity. Especially, dysfunctions in patient allo-immunity against myeloma cells will lead to the acquisition of resistance to therapy. The interaction between PD-L1 on myeloma cells and PD-1 on effector T cells blocks the cytotoxic effects of T cells and high expression of PD-L1 in myeloma cells leads to myeloma cell immune evasion. In melanoma cells, the inhibition of HDAC6 downregulates PD-L1 expression through transcriptional modifications [53]. Further, the transcriptional activation of *PD-L1* is mediated by STAT3 and HDAC6 plays an important role in recruiting STAT3 to the *PD-L1* promoter. The expression of PD-L1 on myeloma cells was shown to be suppressed by treatment with ricolinostat. At the same time, this was found to reduce CD4^+^CD25^+^FoxP3^+^ regulatory T cells and myeloid-derived suppressor cells, leading to enhanced anti-tumor immunity [54]. Furthermore, the expression of PD-L1 on regulatory T cells and PD1 on CD3^+^ T cells is reduced by treatment with ricolinostat. Immune checkpoint molecules including CD80 and CD86, as well as the expression of MHC class I and II, were also found to be enhanced. Moreover, the number of antigen-specific central memory T cells is increased through the activation of AKT/mTOR/p65 via this treatment. Interestingly, the combination of ACY-241, another HDAC6 inhibitor, and anti-PD-L1 enhances MM patient NK cell-mediated cytolytic activity against patient autologous myeloma cells through PD-L1 downregulation on plasmacytoid dendritic cells [55]. Thus, regulation of the immune state by HDAC inhibition has also been reported for the treatment of hematological malignancies.

NKG2D is a major activating immunoreceptor of NK cells and these cells display anti-tumor effects mediated by binding between NKG2D and its ligand, MICA and MICB, on tumor cells. Treating leukemic cells with trichostatin A, an HDAC inhibitor, changes the epigenetic state of *MICA* and *MICB* promoters from a suppressive to an active state and upregulates the expression of these genes [6]. Recently, upregulated expression of MICA in myeloma cells in response to HDAC inhibitor treatment was reported. The cytotoxicity of cytokine-induced killer cells against myeloma cells was also found to be augmented by the combination of three drugs, specifically sodium butyrate (an HDAC inhibitor), matrix metalloproteinase inhibitor III (to block ligand shedding), and phenylarsine oxide (to obstruct surface ligand internalization). This augmented cytotoxicity was suggested to be due to the potent upregulation and stabilization of MICA on the surface of myeloma cells [56]. Moreover, the combination of daratumumab, an anti-CD38 antibody, and dexamethasone with lenalidomide or bortezomib was shown to result in a favorable outcome for relapsing/refractory MM patients [57,58]. However, the expression of CD38, a target of daratumumab, is downregulated soon after its administration and this downregulation is thought to be related to the acquisition of resistance to daratumumab [59]. In myeloma cells, HDAC1 and 2 and IKZF1 and 3, the targets of IMiDs, cooperatively suppress the transcription of *CD38* mRNA, and HDAC inhibitors and IMiDs synergistically upregulate CD38 expression (Figure 3) [60]. Thus, the possibility that HDAC inhibitors could enhance the anti-tumor effects of antibody treatments for myeloma has been supposed.

## 6. Summary and Conclusions

High expression of HDACs including HDAC1 suggests the potential for these enzymes to be exploited as therapeutic targets for the treatment of MM. Their relevance to osteolytic lesions suffered by many MM patients has also been reported. HDACs regulate the expression of many cancer-associated proteins including p21^WAF1^ and RASSF4 through histone modification. These results indicate the rationale for targeting this epigenetic regulatory event using HDAC inhibitors, as treatment strategies for MM. In addition, the HDAC-mediated inhibition of non-histone targets can result in anti-myeloma effects. Accordingly, we discovered PPP3CA as a novel target for MM and revealed that panobinostat suppresses myeloma cell growth through the degradation of this protein, which is released from over-acetylated HSP90. Furthermore, enhancing immune functions to target myeloma cells through the use of HDAC inhibitors will strengthen antibody therapies for the treatment of this disease. Thus, treating MM patients with HDAC inhibitors can be effective through multiple modes of action.

## Figures and Tables

**Figure 1 cancers-11-00475-f001:**
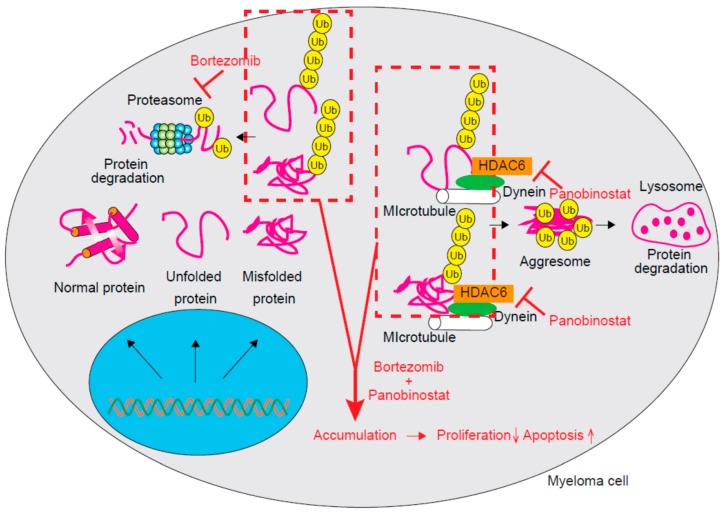
The molecular mechanism underlying the effect of combination therapy comprising bortezomib and panobinostat. Ubiquitinated unfolded/misfolded proteins are degraded by the proteasome. Those proteins escaping from proteasomal degradation will form aggresome complexes with dynein on microtubules and be degraded in the lysosome. The addition of panobinostat will inhibit HDAC6 and block aggresome formation. Accumulated unfolded/misfolded proteins will decrease the proliferation and increase the apoptosis of myeloma cells. Ub, Ubiquitin.

**Figure 2 cancers-11-00475-f002:**
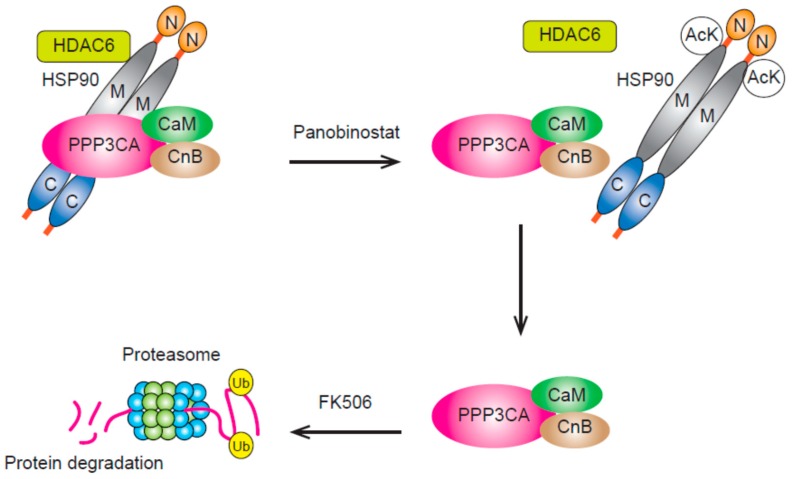
PPP3CA is a client protein of HSP90 and is protected from proteasomal degradation by binding HSP90. The treatment of myeloma cells with panobinostat will result in the over-acetylation of lysines of HSP90 through HDAC6 inhibition. PPP3CA released from HSP90 will be degraded by the proteasome. The interaction between PPP3CA and calcineurin B leads to the protein stability of PPP3CA and the addition of FK506 will enhance its degradation by interfering with the interaction between PPP3CA and calcineurin B. AcK, acetylated lysine; CaM, calmodulin; CnB, calcineurin B; Ub, Ubiquitin; N, amino terminus; M, middle domain; C, carboxy-terminus.

**Figure 3 cancers-11-00475-f003:**
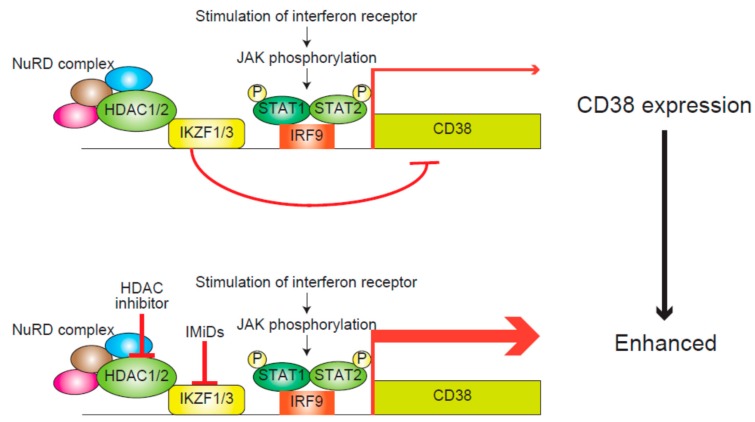
The mechanism underlying the transcriptional regulation of interferon-stimulated genes including CD38 [60]. The interferon-JAK/STAT, IRF9 pathway activates the transcription of interferon-stimulated genes including CD38. In contrast, HDAC1/2 with the NuRD complex synergistically represses those genes in association with IKZF1/3. NuRD, nucleosome remodeling deacetylase; P, phosphorylation.

**Table 1 cancers-11-00475-t001:** The characteristics of each HDAC Classes, substrates, biological functions and IC_50_ values for panobinostat, ricolinostat, and vorinostat with respect to each HDAC (HDAC1, 2, 3, 4, 6, 8, 9) are displayed. IC_50_ values for panobinostat, ricolinostat, and vorinostat was cited from [2,3,4], respectively. GATA1/2, GATA binding factor 1/2; STAT1/3, signal transducers and activators of transcription 1/3; C/EBPα, CCAAT/enhancer binding protein α; SMC3, structural maintenance of chromosome 3.

HDAC	Class	Substrate	Biological Function	IC_50_		
				Panobinostat	Ricolinostat	Vorinostat
HDAC1	I	Histone, p53, GATA1/2, STAT1/3, C/EBPα	Epigenetic regulation Alteration of activity of transcription factors Dysregulation of signaling pathways	≤10 nM	≤100 nM and >10 nM	≤100 nM and >10 nM
HDAC2	I	Histone, GATA1/2, STAT1/3, C/EBPα	Epigenetic regulation Alteration of activity of transcription factors Dysregulation of signaling pathways	≤100 nM and >10 nM	≤100 nM and >10 nM	≤100 nM and >10 nM
HDAC3	I	Histone, GATA1/2, STAT1/3, C/EBPα	Epigenetic regulation Alteration of activity of transcription factors Dysregulation of signaling pathways	≤10 nM	≤100 nM and >10 nM	≤1000 nM and >100 nM
HDAC4	IIa	GATA1/2, STAT1/3, C/EBPα	Epigenetic regulation Alteration of activity of transcription factors Dysregulation of signaling pathways	≤1000 nM and >100 nM	>1000 nM	≤1000 nM and >100 nM
HDAC6	IIb	Tubulin, HSP90	Inhibition of protein degradation Dysregulation of signaling pathways	≤100 nM and >10 nM	≤10 nM	≤1000 nM and >100 nM
HDAC8	I	SMC3	Epigenetic regulation	≤1000 nM and >100 nM	≤100 nM and >10nM	≤1000 nM and >100 nM
HDAC9	IIa	GATA1/2, STAT1/3, C/EBPα	Epigenetic regulation Alteration of activity of transcription factors Dysregulation of signaling pathways	≤10 nM	>1000 nM	≤100 nM and >10 nM

**Table 2 cancers-11-00475-t002:** Summary for combinatory approaches using HDAC inhibitors.

Agent	HDAC Inhibitor to Be Combined with Each Agent	Target Molecule	Reference
IMiDs	entinostat, ricolinostat, panobinostat	c-Myc, HO-1, IRF4, c-Myc	Hideshima et al. 2015 [42], Tang et al. 2018 [45]
Cladribine	entinostat	cyclin D1, E2F-1, p21^waf−1^	Wang et al. 2018 [46]
PI3K inhibitor	panobinostat	RARRES3	Mustafa et al. 2017 [47]
mTOR inhibitors	entinostat	Myc, E2F	Simmons et al. 2017 [50]
5-azacytidine	BG45 (HDAC3-selective inhibitor)	DNMT1	Harada et al. 2017 [51]

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
