# Peer review of "HDAC Inhibitors Exert Anti-Myeloma Effects through Multiple Modes of Action"

_cancers, 2019, doi:10.3390/cancers11040475_

Round 1

Reviewer 1 Report

The manuscript by Imai et al provides an overview of the mechanisms whereby HDAC inhibitors exert anti-myeloma effects. The review is well written and clarify the importance of HDAC inhibitors in MM treatment. It seems to meet the quality for publication. I do not have any specific comments.

Author Response

We really appreciate your kind evaluation of our manuscript. We revised the manuscript according to the reviewers' comments. We  will be happy if you evaluate it again.

Reviewer 2 Report

Imai and colleagues review the role and the therapeutic potential of HDACs in multiple myeloma.

A wealth of preclinical research has been so far performed on HDACs, some of which have been approved for clinical usage in relapsed and refractory myeloma patients.

The topic is interesting, but the authors should improve the organization and the readibility of the review.

Major points:

- English should be revised throught the manuscript possibly by a native speaker;

-The references are not properly linked to the manuscript;

- I would recommend that the authors discuss the role and the therapeutic potential of each HDAC member in a separate paragraph;

-The role of HDAC3, HDAC4, HDAC6, HDAC10 and HDAC11 should be adequately discussed, in terms of: i) expression, ii) molecular mechanisms,iii) prognostic significance and iv)  therapeutic targeting. The part on PP3CA should be reduced and  integrated in such paragraphs.

- The role of HDAC6 in aggresome pathway, and targeting by ricolinostat, should be better discussed, also citing adequate literature (Santo et al.,  Blood 2012; Hideshima et al., PNAS 2016).

- The advamtages of selective HDACi as compared to panHDACi should be bettere stressed.

- The epi-miRNAs targeting HDACs should be adequately discussed, also mentioning recent literature ( Amodio et al., Molecular Cancer Ther 2016; Stamato et al., Oncotarget 2017).

- A table summarizing combinatory approaches with HDAC inhibitors should be provide.

Author Response

1. English should be revised throughout the manuscript possibly by a native speaker;

Our response: Thank you for your suggestion. According to the reviewer 2’s suggestion, we have got the English proofreading by editage (Cactus Communications) for our manuscript.

2. The references are not properly linked to the manuscript;

Our response: Thank you for your suggestion. We profoundly apologize that the references were not properly linked to the manuscript at the first submission. According to the reviewer 2’s suggestion, we reviewed all the references and linked them properly to the manuscript.

3. I would recommend that the authors discuss the role and the therapeutic potential of each HDAC member in a separate paragraph;

Our response: Thank you for your suggestion. According to the reviewer 2’s suggestion, we created the Section 3” The roles of each HDAC in the treatment of MM (expression, prognostic significance, molecular mechanisms, and therapeutic targeting revealed by preclinical studies) ). In this section, we discussed the role and the therapeutic potential of each class (I, IIa, IIb, IV) of HDAC members in a separate paragraph. We also deleted two old sentences, “Targeting epigenetic regulation through histone modification by HDAC inhibitors for the treatment of MM” and “Non-histone proteins can be targeted by HDAC inhibition for the treatment of MM”. 

4. The role of HDAC3, HDAC4, HDAC6, HDAC10 and HDAC11 should be adequately discussed, in terms of: i) expression, ii) molecular mechanisms, iii) prognostic significance and iv) therapeutic targeting. The part on PP3CA should be reduced and integrated in such paragraphs.

Our response: Thank you for your suggestion. According to the reviewer 2’s suggestion, we created the Section 3” The roles of each HDAC in the treatment of MM (expression, prognostic significance, molecular mechanisms, and therapeutic targeting revealed by preclinical studies) ). In this section, we discussed the roles (expression, molecular mechanisms, prognostic significance and therapeutic targeting ) of HDAC3 (class I), HDAC4 (class IIa), HDAC6 (class IIb), HDAC10 (class IIb) and HDAC11(class IV) in a separate paragraph. The part of PPP3CA was reduced and integrated in a paragraph of HDAC6 and HDAC10 (class IIb).

5. The role of HDAC6 in aggresome pathway, and targeting by ricolinostat, should be better discussed, also citing adequate literature (Santo et al., Blood 2012; Hideshima et al., PNAS 2016).

Our response: Thank you for your suggestion. According to the reviewer 2’s suggestion, we added the discussion about the role of HDAC6 in aggresome pathway, and targeting by ricolinostat with the citations of Santo et al., Blood 2012 and Hideshima et al., PNAS 2016 in line 160-167 of revised manuscript.

6. The advantages of selective HDACi as compared to panHDACi should be better stressed.

Our response: Thank you for your suggestion. According to the reviewer 2’s suggestion, we added the description, ” This synergistic activity of HDAC inhibitors with bortezomib is mainly due to HDAC6 and selective HDAC6 inhibitors like ricolinostat and WT161 may reduce the toxicity related to the off-target effects of pan-HDAC inhibitors.”, in line 165-167 of revised manuscript..

7. The epi-miRNAs targeting HDACs should be adequately discussed, also mentioning recent literature (Amodio et al., Molecular Cancer Ther 2016; Stamato et al., Oncotarget 2017).

Our response: Thank you for your suggestion. According to the reviewer 2’s suggestion, we added the description, “Stamato et al. provided novel insights into epigenetic regulation of miR-29b suppression in MM (Stamato et al. 2017). In plasma cells derived from patients with MM, the correlation between miR-29b and EZH2 mRNA expression was inverse. Inhibition of EZH2 lead to miR-29b upregulation through reduced H3K27-trimethylation (H3K27me3) of its promoter regions. Induction of miR-29b thorugh EZH2 inhibition was also found to downregulated major miR-29b pro-survival targets, such as SP1, MCL-1 and CDK6.”, in line 140-144 of revised manuscript. In the paragraph of HDAC4 (class IIa) describing the epi-miRNAs targeting HDACs, we cited both Amodio et al., Molecular Cancer Ther 2016 and Stamato et al., Oncotarget 2017. 

8. A table summarizing combinatory approaches with HDAC inhibitors should be provide.

Our response: Thank you for your suggestion. According to the reviewer 2’s suggestion, we created Table 2 describing “Combinatory approaches with HDAC inhibitors”.

Reviewer 3 Report

The review presented by Lmai et al tried to introduce the effect of HDAC inhibitors in multiple myeloma, which could be of interest to the related researchers. However, due to massively lack of references, non-logical subjective speculation and chaos arrangement of the content, the manuscript cannot be published on cancers.

1. 50% of references are missing from the manuscript.

2. Since the authors are focused on HDACs inhibitors, they should at least list the names of the inhibitors that are used in the clinic, and the preclinical observations of the HDACi. But authors ignored this part but described how to enhance HADCi effect and a potential target for HDACi, which is quite confusing.

3. The logic of part two “Aberrant expression and function of HDACs related to the pathogenesis of MM ”is weird. From one research that showed the high expression of HDAC1 and low expression of HDAC6, the author confirmed, “these results indicate that the overexpression of class I HDACs, and particularly HDAC1, is associated with poor prognosis in MM.” How about the other HDACs, such as HDAC2, 3, 4, 8 and 9? Another non-logical example is “…Although combination therapy with panobinostat and bortezomib is very effective, continuing this therapy becomes difficult due to severe patient fatigue. HDAC3 plays an important role in glucose metabolism muscles. Thus, the strong inhibition of HDAC3 by panobinostat might be related to the severe fatigue associated with panobinostat….(line 159-163)” Just because of HDAC3 related to glucose mechanism in muscles, the author confirmed its relationship to the fatigue, which is not reasonable. Those are just two examples among all the others to show that the authors got the conclusion without careful consideration but in a rash way.

4. Part3 and 4, the author tried to explain the mechanism of HDACi in MM. But from epigenetic modulation, they only showed that HDACi changed the expression level, but failed to explain how did this happen. Since HDACi functions to change the epigenetic modulation, the author should describe how HDACi changed the histone or DNA methylation.

5. Some of the expressions are redundant and non-logical, for example “…Class I HDACs (1, 2, 3, and 8) are generally located in the nucleus and regulate gene expression through the deacetylation of histones. Thus, these have been suggested to function as epigenetic regulators through histone modifications…” (page 2, line 45-46). “…These proteins are sometimes cytotoxic and will interfere with cell function;…” (line 122) .Those expressions are very confusing. Please change into the professional language.

6. Some of the abbreviations come from nowhere that would cause confusion to the readers. Such as MM, ACY-1215 and some others. 

Author Response

1.      50% of references are missing from the manuscript.

Our response: Thank you for your suggestion. We profoundly apologize that the references were not properly linked to the manuscript at the first submission. According to the reviewer 3’s suggestion, we reviewed all the references and linked them properly to the manuscript.

2.      Since the authors are focused on HDACs inhibitors, they should at least list the names of the inhibitors that are used in the clinic, and the preclinical observations of the HDACi. But authors ignored this part but described how to enhance HADCi effect and a potential target for HDACi, which is quite confusing.

Our response: Thank you for your suggestion. According to the reviewer 3’s suggestion, we created Section 2, “HDAC inhibitors in the clinic”, and list the names of the inhibitors that are used in the clinic and described their practical usage. Furthermore, we described the preclinical observations of each HDAC inhibitor in section 3, “The roles of each HDAC in the treatment of MM (expression, prognostic significance, molecular mechanisms, and therapeutic targeting revealed by preclinical studies) )”.

3.      The logic of part two “Aberrant expression and function of HDACs related to the pathogenesis of MM” is weird. From one research that showed the high expression of HDAC1 and low expression of HDAC6, the author confirmed, “these results indicate that the overexpression of class I HDACs, and particularly HDAC1, is associated with poor prognosis in MM.” How about the other HDACs, such as HDAC2, 3, 4, 8 and 9? Another non-logical example is “…Although combination therapy with panobinostat and bortezomib is very effective, continuing this therapy becomes difficult due to severe patient fatigue. HDAC3 plays an important role in glucose metabolism muscles. Thus, the strong inhibition of HDAC3 by panobinostat might be related to the severe fatigue associated with panobinostat…. (line 159-163)” Just because of HDAC3 related to glucose mechanism in muscles, the author confirmed its relationship to the fatigue, which is not reasonable. Those are just two examples among all the others to show that the authors got the conclusion without careful consideration but in a rash way.

Our response: Thank you for your suggestion. According to the reviewer 3’s suggestion, we described clinical significance of expression of each HDAC in the Section 3” The roles of each HDAC in the treatment of MM (expression, prognostic significance, molecular mechanisms, and therapeutic targeting revealed by preclinical studies) ). The part of “the strong inhibition of HDAC3 by panobinostat might be related to the severe fatigue associated with panobinostat”

was deleted.

4.      Part3 and 4, the author tried to explain the mechanism of HDACi in MM. But from epigenetic modulation, they only showed that HDACi changed the expression level, but failed to explain how did this happen. Since HDACi functions to change the epigenetic modulation, the author should describe how HDACi changed the histone or DNA methylation.

Our response: Thank you for your suggestion. According to the reviewer 3’s suggestion, we added the description, “One of the main biological functions of class I HDACs is an epigenetic regulation (Table 1). The pattern of DNA methylation and posttranslational modifications of histone regulate epigenome (Thiagalingam et al. 2003). Acetylation of histones and methylation at the lysine 9 residue of histone 3 lead to the activated gene expression. In contrast, the suppression of gene expression is associated with histone deacetylation by HDACs frequently associated with regions of DNA methylation. Inappropriate silencing of tumor suppressor genes may be related to the occurrence of various type of cancers. HDAC inhibitors cause the accumulation of acetylated histones in nucleosomes. The hydroxamic acid parts of HDAC inhibitors bind to the zinc in the tubular pocket of HDACs and this interaction is suppose to inhibit the catalytic activity of HDACs.” in line 94-101 of revised manuscript.

5.      Some of the expressions are redundant and non-logical, for example “…Class I HDACs (1, 2, 3, and 8) are generally located in the nucleus and regulate gene expression through the deacetylation of histones. Thus, these have been suggested to function as epigenetic regulators through histone modifications…” (page 2, line 45-46). “…These proteins are sometimes cytotoxic and will interfere with cell function; …” (line 122) .Those expressions are very confusing. Please change into the professional language.

Our response: Thank you for your suggestion. According to the reviewer 3’s suggestion, we have got the English proofreading by editage (Cactus Communications) for our manuscript.

6. Some of the abbreviations come from nowhere that would cause confusion to the readers. Such as MM, ACY-1215 and some others. 

Our response: Thank you for your suggestion. According to the reviewer 3’s suggestion, we reviewed all the abbreviations and new-added abbreviations were demonstrated by red-colored text.

Reviewer 4 Report

The review of Imai Yoichi and colleagues summarized the role of HDAC inhibitors as anti-myeloma drug in combination with others molecules. The remarkable use of HDACIs in experimental studies in Multiple Myeloma is confirmed by the high number of original papers or reviews recently published (Monoclonal Antibodies versus Histone Deacetylase Inhibitors in Combination with Bortezomib or Lenalidomide plus Dexamethasone for the Treatment of Relapsed or Refractory Multiple Myeloma: An Indirect-Comparison Meta-Analysis of Randomized Controlled Trials. Zheng Y J Immunol Res. 2018 Jun 27, Novel agents in the treatment of multiple myeloma: a review about the future. Naymagon L, J Hematol Oncol. 2016 Jun 30;9(1):52; New insights into the treatment of multiple myeloma with histone deacetylase inhibitors. Cea M, Curr Pharm Des. 2013;19(4):734-44. Review).

Despite being a very interesting topic, the work lacks of many things that should be done to publish it.

1-      Table 1: this table is not clear, the authors introduce the IC50 value indicated with symbol (+ or -) referred to single HDACs inhibition exerted by different  HDACI, they should better explain that this enzymatic activity inhibition; moreover, where they got this data? No references present.. please add

2-      In the Figure 1, the authors described the molecular mechanism of HDACIs/bortezomib combination. We suggest to better explain the two pathways: proteasome and aggresome…

3-      Imai Yoichi and colleagues described several target proteins involved in the combination strategies but frequently the authors don’t describe their functions (i.e. EZH2, TS, DNMT, CRBN, B-CCL, CAM-DR…), please define them.

4-      A lot of references were not reported (for example line 60) or not corrected insert in the text (for example line 151, 153, 164): we suggest to review all the references

5-      Please revised the name of some cited HDACIs in the text, for example (vorinostat /SAHA, line 96), entinostat/MS275 (line 180), please use the same words.

6-      Please revised the name of the tumor suppressor RASSF4 (line 106-108)  and correct in line  104 that  is defined as  tumor suppressive??? , please use the same words.

7-      We suggest to review extensively all the text for English, 

Author Response

1.      Table 1: this table is not clear, the authors introduce the IC50 value indicated with symbol (+ or -) referred to single HDACs inhibition exerted by different HDACI, they should better explain that this enzymatic activity inhibition; moreover, where they got this data? No references present. please add

Our response: Thank you for your suggestion. According to the reviewer 4’s suggestion, we modified the legend of Table 1. We also added the information of references.

2.      In the Figure 1, the authors described the molecular mechanism of HDACIs/bortezomib combination. We suggest to better explain the two pathways: proteasome and aggresome…

Our response: Thank you for your suggestion. According to the reviewer 4’s suggestion, we added the description, “Panobinostat and its combination with bortezomib can synergistically induce apoptosis by inhibiting protein degradation and massive accumulation of polyubiquitinated proteins by targeting both proteasomal and aggresomal protein degradation systems (Figure 1) (Hideshima, Richardson and Anderson 2011).” in line 162-165 of revised manuscript.

3.      Imai Yoichi and colleagues described several target proteins involved in the combination strategies but frequently the authors don’t describe their functions (i.e. EZH2, TS, DNMT, CRBN, B-CCL, CAM-DR…), please define them.

Our response: Thank you for your suggestion. According to the reviewer 4’s suggestion, we added the descriptions, “EZH2 regulates cell proliferation in the hematopoietic cells;” for EZH2 in line 121-122,

“tumor suppressor genes like p15 (INK4b) and ESR1” for TS in line 131,

“DNMTs methylates DNA at the carbon-5 position of cytosines mainly within CpG sites as epigenetic process involved in the regulation of gene expres­sion (Gao et al. 2018). Preferential methylation by DNMT1 is performed at hemi-methylated CpG sites to maintain DNA methylation during DNA replication and cell division.” for DNMT in line 344-347,

“CRBN is an IMiDs’ binding protein and the substrate adaptor of the CRL4CRBN E3 ubiquitin ligase (Ito et al. 2010), (Fink and Ebert 2015). IMiDs induce the recruitment of specific substrates including IKZF1/3 to E3 ubiquitin ligase and display anti-myeloma effects through the ubiquitination and subsequent proteasomal degradation of IKZF1/3.” for CRBN in line 288-291,

“B-CLL, a mature B-cell malignancy” for B-CCL in line 313 of revised manuscript.

We deleted the part including CAM-DR according to the reviewer 2’s comment 4.

4.      A lot of references were not reported (for example line 60) or not corrected insert in the text (for example line 151, 153, 164): we suggest to review all the references

Our response: Thank you for your suggestion. We profoundly apologize that the references were not properly linked to the manuscript at the first submission. According to the reviewer 4’s suggestion, we reviewed all the references and linked them properly to the manuscript.

5.      Please revised the name of some cited HDACIs in the text, for example (vorinostat /SAHA, line 96), entinostat/MS275 (line 180), please use the same words.

Our response: Thank you for your suggestion. According to the reviewer 4’s suggestion, we unified the name of each HDAC inhibitor and the revised names were demonstrated by red-colored text.

6.      Please revised the name of the tumor suppressor RASSF4 (line 106-108) and correct in line 104 that is defined as tumor suppressive??? , please use the same words.

Our response: Thank you for your suggestion. According to the reviewer 4’s suggestion, we corrected the description and the change was demonstrated by red-colored text.

7.      We suggest to review extensively all the text for English, 

Our response: Thank you for your suggestion. According to the reviewer 4’s suggestion, we have got the English proofreading by editage (Cactus Communications) for our manuscript.

Round 2

Reviewer 2 Report

Although major points raised by reviewers have been addressed, there are some mistakes which MUST be corrected.

Line 91, pag 3:

Authors state that one of the main biological functions of class I HDACs is an epigenetic regulation. 

Note that this is a feature of all HDACs!

Pag 7, line 221: the authors state that HDAC11 is an HDAC inhibitor rather than a protein!!!!!

Legends need to be corrected. For instance in Table 1: "the character"  of each HDAC, please correct.

The content of the table is of difficult interpretation, especially for the IC50 (+++, ++, + cannot be easily understandable).

Author Response

We appreciate the comments and suggestions from the reviewers who contributed to improving our manuscript. We have revised the manuscript to address the comments of the reviewers, and have included a description of our point-by-point responses to the reviewers’ comments as follow. The changes to our manuscript are demonstrated by orange-colored text.

Our point-by-point responses to the reviewers’ comments

The reviewer 2’s comments

(1)   Line 91, pag 3: Authors state that one of the main biological functions of class I HDACs is an epigenetic regulation. Note that this is a feature of all HDACs!

Our response: Thank you for your suggestion. We change the description to “Epigenetic regulation is one of the main biological functions of HDACs including class I HDACs (Table 1)”.

(2)   Pag 7, line 221: the authors state that HDAC11 is an HDAC inhibitor rather than a protein!!!!!

Our response: Thank you for your suggestion and we apologize for our mistake. According to the reviewer 2’s suggestion, we changed the description to “HDAC11 is the smallest HDAC as well as only one member of class IV HDAC”.

(3)   Legends need to be corrected. For instance in Table 1: "the character" of each HDAC, please correct.

Our response: Thank you for your suggestion. According to the reviewer 2’s suggestion, we changed the descriptions to those as follow.

Legend of Table 1: to “The characteristics of each HDAC”

Legend of Table 2: to “Summary for combinatory approaches using HDAC inhibitors”

(4)   The content of the table is of difficult interpretation, especially for the IC50 (+++, ++, + cannot be easily understandable).

Our response: Thank you for your suggestion. According to the reviewer 2’s suggestion, we changed the descriptions to those as follow.

Table 1

We changes the desctiprions of IC50 to those as follow. From “-“ to “ > 1000 nM”, “+” to “£ 1000 nM and > 100 nM”, “++” to “£100 nM and > 10 nM”, “+++” to “£ 10 nM”. We also deleted “IC50; -: > 1000 nM, +: £ 1000 nM and > 100 nM, ++: £100 nM and > 10 nM, +++: £ 10 nM” in the legend of Table 1. Then, we added “IC50 values for panobinostat, ricolinostat, and vorinostat was cited from [2], [3], and [4], respectively” there.

Table 2

The 1st line and 2nd row of Table 2: from “HDAC inhibitor to be combined” to “HDAC inhibitor to be combined with each agent”

(5)   According to the reviewer 2’s suggestion, we have got the English proofreading by editage (Cactus Communications) for our manuscript again. The changes to our manuscript are demonstrated by orange-colored text.

(6)   According to the editor’s suggestion, we put Tables 1 and 2 after their first citation and added the refference numbers for all refferences in the Table 2. We added “The described strategy for enhancing the anti-myeloma effect of HDAC inhibitors by adding other agents is summarized in Table 2.” in line 315-316.

Reviewer 3 Report

The author addressed most of my questions, I do not have further questions.

Author Response

We appreciate the comments and suggestions from the reviewers who contributed to improving our manuscript. We have revised the manuscript to address the comments of the reviewers, and have included a description of our point-by-point responses to the reviewers’ comments as follow. The changes to our manuscript are demonstrated by orange-colored text.

Thank you for your evaluation of our manuscript. We revised the manuscript according to the reviewer 2’s suggestion. Furthermore, we have got the English proofreading by editage (Cactus Communications) for our manuscript again. The changes to our manuscript are demonstrated by orange-colored text.

Reviewer 4 Report

it is fine for me in the present form 

Author Response

(The authors gave the same response as above.)
